

# Germination pretreatments to break hard-seed dormancy in *Astragalus cicer* L. (Fabaceae)

Joseph M. Statwick[1,2]

[1] Department of Biological Sciences, University of Denver, Denver, Colorado, United States
[2] Department of Research and Conservation, Denver Botanic Gardens, Denver, Colorado, United States

## ABSTRACT

Conservationists often propagate rare species to improve their long-term population viability. However, seed dormancy can make propagation efforts challenging by substantially lowering seed germination. Here I statistically compare several pretreatment options for seeds of *Astragalus cicer* L.: unscarified controls and scarification via physical damage, hot water, acid, and hydrogen peroxide. Although only 30% of unscarified seeds germinated, just physical scarification significantly improved germination, whereas one treatment, hot water, resulted in no germination at all. I recommend that rare species of *Astragalus*, as well as other hard-seeded legumes, be pretreated using physical scarification. Other methods may require considerable optimization, wasting precious time and seeds.

## INTRODUCTION

Propagating wild species in greenhouses and common gardens for their restoration or reintroduction in native habitats can be an effective method of improving the size and viability of rare or threatened populations (*Maunder, 1992*; *Menges, 2008*). Such in situ and ex situ propagation techniques are beneficial, so long as these techniques are successful in establishing additional reproductive adults in novel, degraded, or extirpated sites (*Maunder, 1992*; *Menges, 2008*). If, however, reintroduction is unsuccessful (which it usually is *Godefroid et al. (2011)*), it accomplishes nothing more than wasting resources and even further threatening the species by removing seeds that would have become the future seed bank.

At ~3,270 species, *Astragalus* (Fabaceae) is the largest genus of flowering plants in the world (*Watrous & Cane, 2011*). Though a few *Astragalus* are weedy, wide-ranging generalists, specialization on uncommon and infertile soils seems to be a hallmark of the genus (*Barneby, 1964*). Unfortunately, this specialization appears to restrict many species to small geographic ranges, making them more vulnerable to extinction. In the United States alone, the *US Fish and Wildlife Service (2014)* has listed five *Astragalus* species as threatened and 16 as endangered, with an additional five as candidates for listing, and three more currently under review. Although the International Union for the

Corresponding author
Joseph M. Statwick,
jstatwick@gmail.com

Conservation of Nature (IUCN) red list (*2014*), a global database to track at-risk species, contains less than one half of one percent of known *Astragalus* species, nearly 40 percent of those with sufficient data are considered "vulnerable" or worse (nine vulnerable, 12 endangered, 18 critically endangered, and one extinct). *NatureServe (2014)*, meanwhile, lists 100 vulnerable, 58 imperiled, and 31 critically imperiled species, which combine to nearly a third of the 616 *Astragalus* species in its database.

*Astragalus* species, like most temperate legumes, as well as species of as many as 15 different plant families, have hard seed coats and physical dormancy, which often require scarification or stratification to break (*Baskin, Baskin & Li, 2000*; *Long et al., 2012*). In particular, low germination rate has been observed for several rare species of *Astragalus*, including *A. nitidiflorus* (*Vicente et al., 2011*), *A. bibullatus* (*Albrecht & Penzagos, 2012*), and *A. arpilobus* (*Long et al., 2012*). Physical dormancy is generally adaptive; it helps delay seedling emergence until favorable environmental conditions, particularly in habitats with high seasonal or interannual variation (*Baskin, Baskin & Li, 2000*). Prolonged dormancy of the seed bank may also contribute to the maintenance of genetic diversity in rare *Astragalus* such as *A. albens* by resurrecting extirpated genotypes (*Neel, 2008*). However, this dormancy is counterproductive for ex situ propagation efforts.

Many scarification treatments for various *Astragalus* species have been explored in the literature, including dry heat (*Albrecht & Penzagos, 2012*; *Chou, Cox & Wester, 2012*; *Long et al., 2012*), wet heat (*Acharya et al., 2006*; *Long et al., 2012*), stratification (*Acharya et al., 2006*; *Albrecht & Penzagos, 2012*; *Long et al., 2012*), physical scarification (*Miklas, Townsend & Ladd, 1987*; *Acharya et al., 2006*; *Albrecht & Penzagos, 2012*), acid (*Miklas, Townsend & Ladd, 1987*; *Acharya et al., 2006*; *Long et al., 2012*) smoke water (*Chou, Cox & Wester, 2012*), etc.

Generally, physical scarification tends to be reliable for *Astragalus*, but other treatments been successful in some circumstances (*Acharya et al., 2006*). *Long et al. (2012)* found that the germination of *Astragalus arpilobus* by hot water scarification was maximized at 100 °C for 10 min of exposure, yet no amount of time at 90 °C or below was sufficient to increase germination significantly beyond controls. Fresh *Astragalus cicer* seeds, meanwhile, had maximum germination rates at $\geq$ 15 rounds of alternating liquid nitrogen (−196 °C for 5 min) and steam (100 °C for 5 min). *Astragalus* seeds treated with concentrated sulfuric acid (18 M) for 20 min have been shown to germinate very successfully (*Miklas, Townsend & Ladd, 1987*). Hydrogen peroxide, which is cheaper and safer to use than acid, has been shown to marginally improve the germination of *Ribes cereum* (Rosaceae) (*Rosner et al., 2003*), but does not appear to have been tested in the literature for *Astagalus*.

Despite these successes, it is rare that the results of more than one or two treatments on *Astragalus seeds* have been compared in the same study. Furthermore, because different species and even collections within species vary in germination rate, (*Miklas, Townsend & Ladd, 1987*; *Acharya et al., 2006*; *Albrecht & Penzagos, 2012*), the results of these studies are not directly comparable to one another in order to determine the most effective scarification treatment. I therefore explored five different pre-planting seed treatments to determine which would best promote germination in the generalist forage crop, *Astragalus cicer* "Oxley."

## METHODS

*Astragalus cicer* L. (cicer milkvetch) is an old-world native that was introduced to North America as a hardy, palatable forage crop (*Acharya et al., 2006*). "Oxley" is an ecotype that was first collected in the former USSR and introduced to the United States in 1971 (*Acharya et al., 2006*). Although *A. cicer* is not rare, it is a suitable model for rare species because it is readily commercially available without threatening wild populations, and because it, like its rare congenerics, is well known for its slow stand establishment, largely due to low germination rates and prolonged seed dormancy (*Acharya et al., 2006*).

I exposed a total of 250 *A. cicer* seeds (Granite Seed, Denver, CO, USA) to each of five different scarification treatments (n = 50), starting March 15, 2013 at Denver Botanic Gardens (DBG) in Denver, Colorado. The scarification treatments were physical damage, hot water, hydrogen peroxide, acid, and a control. Control seeds were planted in 1 cm$^2$ cells in a plastic germination tray, without scarification, on the surface of a seed starter mix, and covered with approximately 3 mm of vermiculite. Treated seeds were planted in the same manner, after scarification, in the same 288-cell tray as the control seeds.

For the physical scarification treatment, I cut the seed coat opposite the radicle with a pair of infant nail clippers, being careful to not damage the endosperm or embryo. Because my experiment was performed at ~1,600 m altitude where water boils at < 95 °C, I felt the hot water treatment would require a more prolonged soak than is typical. Thus, the seeds were placed in a thermos of boiling water, covered, and soaked for 20 h before planting. The peroxide treated seeds were soaked in pure ZeroTol, a commercial greenhouse fungicide/algaecide, (27% hydrogen peroxide, BioSafe Systems, East Hartford, CT, USA) for one hour before planting, I chose a more concentrated solution for a shorter duration than was effective for *Ribes* (4–8 h soak in 3% hydrogen peroxide) because of the thicker, more recalcitrant seed coat in legumes and the increasing seed rot observed with longer exposure times (*Rosner et al., 2003*). Acid treated seeds were soaked in lab grade sulfuric acid (98%, 18 M) for five minutes. This is a reduced duration compared to previous studies because at least some seeds were rendered non-viable by acid treatment, although admittedly "very few" (*Miklas, Townsend & Ladd, 1987*).

All seedlings were reared in a propagation greenhouse at DBG. The total number of seeds germinated in each treatment was recorded approximately twice per week for one month. The potting soil was checked daily and kept evenly moist by DBG horticulture staff. Plants were exposed only to natural sunlight, which, given the date and latitude, ranged between approximately 12 h at the beginning of the trial and 13 h and a half hours at the end of the trial.

Germination data were analyzed with a Cox proportional hazards analysis using JMP v10 (SAS, Cary, NC, USA). This analysis type is well suited to germination data in that it is intended for time series datasets composed of binary data in which each observation is a replicate (i.e. each seed has germinated or not germinated), and compares observed and expected frequencies with a $\chi$ distribution. It calculates a pairwise risk ratio (RR) between treatments, where a RR greater than one means higher relative germination and a RR less than one means lower relative germination. Alternatively,

the RR can be interpreted as the likelihood that a random individual from one treatment will reach the endpoint (i.e. germinate) before a random individual from the other treatment (*Spruance et al., 2004*). Seeds that did not germinate during the entire treatment period were right-censored, while all other individuals were interval censored. The statistical significance of post-hoc comparisons was assessed at a Bonferroni-corrected alpha of 0.005. Repeated measures ANOVA was not used because calculating the variance of proportions based on grouped binary data is inappropriate in that the proportions are both ordinal and bounded between 0 and 1.

## RESULTS

Seed treatment was an exceptionally strong predictor of seed germination success ($\chi^2 = 67.6$, P < 0.0001, df = 4, n = 250). Physically scarified seeds germinated most quickly, and were more than twice as successful as any other treatment (Table 1), with a final germination rate of 74% over 33 days (Fig. 1). Statistically similar percentages of unscarified, acid scarified, and peroxide scarified seeds germinated (30, 34, and 26%, respectively) (Table 1). No seeds from the hot water scarification treatment germinated. Across all treatments, the bulk of germination occurred within the first two weeks, with virtually no germination after that point (Fig. 1).

## DISCUSSION

Although many scarification treatments have been attempted for *Astragalus* species, my data show that not all treatments are equal in efficacy. In fact, only one treatment, physical scarification, was significantly better than the control, and the hot water treatment was significantly worse than the control, resulting in no germination at all.

Based on my data, I recommend that propagation efforts involving rare *Astragalus* species use physical scarification as the primary method for breaking seed dormancy. The major disadvantage of using physical scarification, the labor-intensive nature of individually damaging the seed coat with sandpaper, a razor blade, or nail clippers, can be overcome with batch scarification methods. These include abrasive-lined drums or vane polishers for relatively small lots, or commercial seed polishing, hulling, or scarifying equipment for larger lots, albeit at the cost of slightly greater seed loss from damage (*Acharya et al., 2006*). However, the 10/10 rule of wild seed collection (take no more than 10% of the seeds from no more than 10% of the reproductive plants) (*Guerrant et al., 2013*) severely limits the number of seeds available from rare species, which may have only dozens or hundreds of reproductive individuals within a given year. Because seed numbers from these collections are likely limited to the hundreds, the time required to scarify individually is minimal, whereas the higher seed loss with batch scarification equipment would be unacceptable. If individual scarification is impractical because a species is more common or has been propagated ex situ, I suggest performing additional optimization trials specific to the type and model of scarification equipment, according to the manufacturer recommendations.

Although *Astragalus cicer* is a relative generalist that would likely not require the sorts of atypical scarification techniques that might be necessary for strongly specialized

**Table 1 Pairwise risk ratios for treatments, expressed as the ratio of the germination success of the row relative to the column.** For example, the risk ratio of controls relative to nail clippers was 0.32 (32% as likely to germinate), while the risk ratio of nail clippers relative to controls was 3.17 (317% more likely to germinate), n = 50 for each treatment.

| Treatments | Control | Hot water | Sulfuric acid | Nail clippers | Hydrogen peroxide |
|---|---|---|---|---|---|
| Control | 1 | > 100* | 0.85 | 0.32* | 1.17 |
| Hot water | < 0.01* | 1 | < 0.01* | < 0.01* | < 0.01* |
| Sulfuric acid | 1.17 | > 100* | 1 | 0.37* | 1.38 |
| Nail clippers | 3.17* | > 100* | 2.69* | 1 | 3.72* |
| Hydrogen peroxide | 0.85 | > 100* | 0.72 | 0.27* | 1 |

Note:
* Represent statistical significance at the P < 0.001 level. All other post-hoc comparisons were not significant.

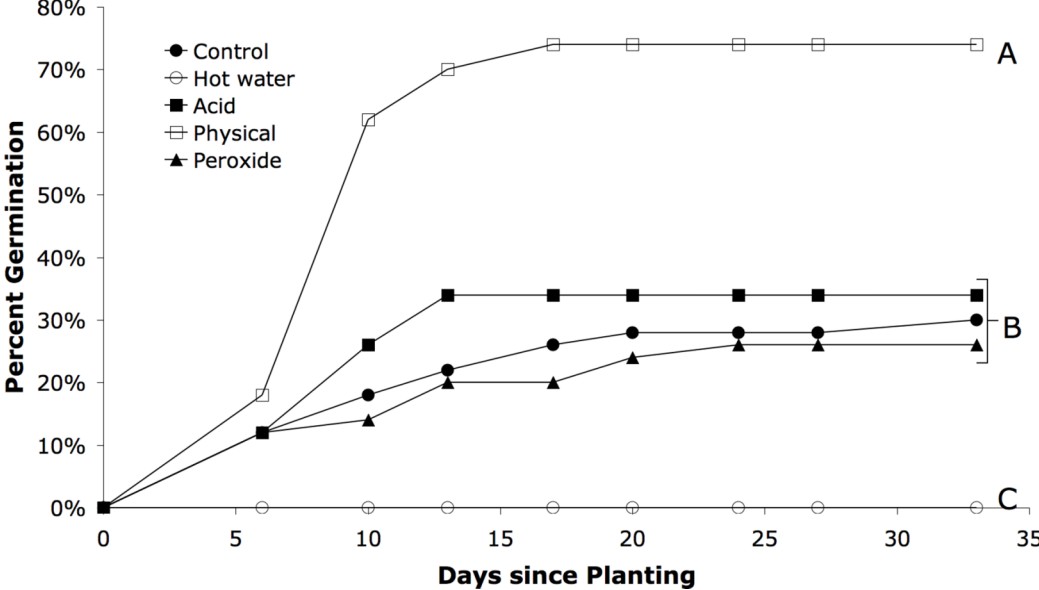

**Figure 1 Germination rates over time for different scarification treatments for _Astragalus cicer_.** Letters indicate statistically different treatments via proportional hazards analysis.

lineages, to my knowledge, there are no reports of physical scarification being ineffective in _Astragalus_. Nonetheless, _Astragalus_ as a genus has a very broad range of morphological and physiological variation, with species that are annual or perennial, endemic to mineral or humic soils, etc. Thus, care should be taken in extending these results across the entire range of _Astragalus_ species.

Still, whereas other studies have demonstrated that methods involving cold, heat, acid, etc., can improve germination over controls, I recommend against their widespread use in _Astragalus,_ as the studies comparing different durations and intensities (i.e. temperature and concentration) of these treatments have found a relatively narrow range of optimal conditions (_Albrecht & Penzagos, 2012_; _Chou, Cox & Wester, 2012_; _Long et al., 2012_). Treatments of insufficient duration or intensity appear to be incapable of breaking seed dormancy, whereas treatments of excessive duration or intensity damage not only the seed coat, but the embryo as well, causing a loss of

viability (*Albrecht & Penzagos, 2012*; *Chou, Cox & Wester, 2012*; *Long et al., 2012*). Even when such treatments are better than controls, I have found no reported instance for an *Astragalus* species in which they are more effective than physical scarification, and they are sometimes still worse (*Miklas, Townsend & Ladd, 1987*; *Acharya et al., 2006*). In addition, some treatments, particularly those that involve concentrated acid, liquid nitrogen, fire, or other reactive substances, could be hazardous and are best avoided unless absolutely necessary.

## CONCLUSIONS

Physical scarification is a simple, safe, and reliable way to improve germination rates in *Astragalus* species with hard seed dormancy. I advise that, particularly for rare species for which seeds are limited, attempting to optimize other techniques is an unnecessary waste of resources unless physical scarification has been demonstrated to be ineffective.

## ACKNOWLEDGEMENTS

I would like to thank Anna A. Sher for her help with data analysis and Jennifer Neale for her comments on the manuscript and facilitation of use of the DBG greenhouse. I also thank the DBG horticulture staff, particularly Mike Bone and Katy Wilcox, for permission to use valuable greenhouse space and planting materials, and for their invaluable aid and expertise. I thank Elizabeth Pilon-Smits for advice on cultivating *Astragalus* species.

### Funding
The author received no funding for this work.

### Competing Interests
The author declares that he has no competing interests.

### Author Contributions
- Joseph M. Statwick conceived and designed the experiments, performed the experiments, analyzed the data, contributed reagents/materials/analysis tools, wrote the paper, prepared figures and/or tables, reviewed drafts of the paper.

### Data Deposition
    The raw data has been supplied as Supplemental Dataset Files.

### Supplemental Information
Supplemental information for this article can be found online at http://dx.doi.org/10.7717/peerj.2621#supplemental-information.

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
