# Peer review of "Germination pretreatments to break hard-seed dormancy in Astragalus cicer L. (Fabaceae)"

_PeerJ, doi:10.7717/peerj.2621_

## Round 0.1 · original submission · Minor Revisions

This was an interesting paper and there is no doubt that identifying efficient, economical methods to propagate native species is an urgent need in restoration ecology. It was nice to see a number of different treatments being compared. That being said there are a number of issues which need your attention (see reviewers comments and the attached, annotated manuscripts). In particular three major areas need to be improved:

- You need to provide some form of justification for the scarification methods you choose especially as some of them seem surprisingly severe. As reviewer 2 points out you also need to ensure your methods are reproducable.

- You need to provide some additional background on research that has assessed the germination process in Astragalus spp.
- Reporting of your statistical methods requires more information with regards to the post-hoc pairwise comparisons you completed.

I would urge you to pay particular attention to my comments on the methods section of your paper. I do have some concerns regarding the thinking behind your chosen treatments, what the justification for them was, whether they could reasonably be expected to improve germination and whether they effectively mimic techniques that could be used on a commercial scale. Though I've made a decision of minor corrections here if these issues cannot me adequately addressed then I would have real hesitation in accepting this paper.

·

Basic reporting

This article is clear, concise and well written. Sufficient background is presented to justify the research. The figure would be improved by including a legend instead of the lengthy verbal description.

Experimental design

Simple and straightforward, this design adequately addresses the research question. N of 50 is appropriate for germination trials with recalcitrant seeds. Scarification methods are adequately described and repeatable. The question being addressed is indeed of interest to the broader community.

Validity of the findings

Data is available, clear and sufficiently robust. Recommendations about the genus Astragalus as a whole may reach beyond the scope of this experiment, although this work certainly would inform those attempting to germinate other species of Astragalus. In my own work trialing scarification methods on hard-coated legumes (Lupinus spp.), scarification methods that are effective for one species are not necessarily effective for others in the same genus.
Physical scarification was effective in this experiment as individual attention was given to each of 50 seeds to ensure that the seed coat was broken. While this is applicable in a limited research project, this scarification method is not practical on a landscape restoration scale. The author does suggest that batch scarification is possible with mechanical equipment; however this would require similar “optimization” trials that were rejected for other scarification methods in this study. Again from personal experience I agree that physical scarification is the preferred method for hard coated legumes, but not necessarily for the reasons presented by the author.

·

Basic reporting

The background should be expanded to clarify what is known about dormancy breaking treatments in Astragalus species, specifically.

Experimental design

The knowledge gap should be clarified in relation to dormancy breaking treatments previously explored in Astagalus (or, if unavailable, in other hard-seeded species - in which case, these should be named).
The basis for the specific treatment protocols should be explained - for example, why did the author choose to place the seeds in a thermos of boiling water (etc.)? It is unclear how the treatments build on previous research.
The author should provide estimates of exposure time and temperature for the hot water and fire treatments so that they can be reproduced.
The statistical analysis of the data shown in Table 1 should be provided. Information on how to access the software used to analyze the data in Figure 1 should be provided.

Validity of the findings

The author should provide the basis for concluding that seeds either remained dormant (e.g., the acid treatment) or were rendered nonviable (the hot water and fire treatments). It's not clear how this was determined.
The basis for the concluding sentence is not clear (see my marginal comment).
Is it really appropriate to apply your results to all Astragalus species? Might not some species have specialized dormancy breaking requirements? Some hard-seeded species have multiple dormancy mechanisms.

---

## Round 0.2 · Minor Revisions

Thanks for your hard work attending to the suggested revisions and for your detailed response to the reviewers comments. My apologies for the delay in this decision - I've been away on fieldwork without internet access for the last few weeks.

In general I feel the paper is much improved from the previous version that being said I have a couple of remaining suggestions and one more major concern:

MINOR SUGGESTIONS
You've added a lot of justification for the treatments you've used. This is mostly found in the Methodology I wonder if this background would be better presented in the Introduction to provide a more complete overview of relevant previous research?

Paragragph 2 of the methods reads like you treated a total of 50 seeds across all treatments whereas you actually used 50 seeds per treatment - please clarify

CONCERNS
I have significant concerns about the fire treatment used in this experiment:
1) I can't agree with your statement regarding fire temperatures and durations (30 minutes at 125C) - this is a huge generalisation as heat exposure (temperature residence) will vary hugely between ecosystems
(e.g. forest versus grassland fires) and on the position of the seed in the soil - heating declines exponentially with soil depth. Most vegetation fires have residence times from a few seconds to a few minutes. In reasonable moist soils in temperate climates only increases by a few tens of degrees at most. Thirty minutes at 125 would indicate prolonged smouldering of heavy fuels (downed woody debris nearby or above
the location). Its unlikely such conditions high severity conditions would be associated with low fire intensity. You should consult a wider variety of fire ecology literature looking at soil heating during vegetation fires and its effect on seed germination.
2) As your seeds were located on the soil surface and covered with dry needles they were essentially part of the fuel bed and it is unsuprising that they were no longer viable. I'm very skeptical of this treatment as it lacks control or replicability and there is no information provided on the fuel load, burn time or heating associated with the treatments (30-60 seconds is a huge range). I think this particular treatment is extremely flawed. The justification of broadcast seeding followed by prescribed burning is nonsensical from a management perspective as that would likely lead to a significant amount of seed mortality and a waste of money. Can you point to examples of such an approach actually being used? A more realistic option would be dry heat pre-treatment of seeds prior to broadcasting.

In view of the above I think the simplest solution would be to remove this treatment from your analysis.

Finally I would ask that you more fully address the issue raised in the first round of review - that the physical scarification method is not directly comparable with the other "batch" methods. It is unclear why you did not attempt a batch physical scarification method and this should be explained.

·

Basic reporting

No comments

Experimental design

No Comments

Validity of the findings

No comments

Additional comments

The author has adequately addressed my concerns with the manuscript.

---

## Round 0.3 · accepted · Accept

Thanks for making the requested changes. I've now accepted the paper. With regards to the fire treatment, during review of proofs, you may wish to insert a line suggesting this as an avenue for further research.